# Marked-LIEO: Visual Marker-Aided LiDAR/IMU/Encoder Integrated Odometry

**DOI:** 10.3390/s22134749

**Published:** 2022-06-23

**Authors:** Baifan Chen, Haowu Zhao, Ruyi Zhu, Yemin Hu

**Affiliations:** 1School of Automation, Central South University, Changsha 410017, China; chenbaifan@csu.edu.cn (B.C.); zhaohaowu@csu.edu.cn (H.Z.); ruee1101@csu.edu.cn (R.Z.); 2School of Resources and Safety Engineering, Central South University, Changsha 410017, China

**Keywords:** integrated odometry, pre-integration, visual marker, multi-sensor fusion

## Abstract

In this paper, we propose a visual marker-aided LiDAR/IMU/encoder integrated odometry, Marked-LIEO, to achieve pose estimation of mobile robots in an indoor long corridor environment. In the first stage, we design the pre-integration model of encoder and IMU respectively to realize the pose estimation combined with the pose estimation from the second stage providing prediction for the LiDAR odometry. In the second stage, we design low-frequency visual marker odometry, which is optimized jointly with LiDAR odometry to obtain the final pose estimation. In view of the wheel slipping and LiDAR degradation problems, we design an algorithm that can make the optimization weight of encoder odometry and LiDAR odometry adjust adaptively according to yaw angle and LiDAR degradation distance respectively. Finally, we realize the multi-sensor fusion localization through joint optimization of an encoder, IMU, LiDAR, and camera measurement information. Aiming at the problems of GNSS information loss and LiDAR degradation in indoor corridor environment, this method introduces the state prediction information of encoder and IMU and the absolute observation information of visual marker to achieve the accurate pose of indoor corridor environment, which has been verified by experiments in Gazebo simulation environment and real environment.

## 1. Introduction

Simultaneous localization and mapping is the premise for mobile robots to carry out automatic navigation tasks in the indoor unknown environment. Localization information is particularly critical, which provides important prior information for the path planning, feedback control, and emergency obstacle avoidance of mobile robots. SLAM system with high robustness, high precision, and strong real-time performance has always been the focus of related research. The realization of the SLAM system mainly includes two categories: vision-based methods and LiDAR-based methods. Vision-based methods usually use a monocular camera, stereo camera, or RGBD camera to estimate the motion of the camera. The most classic visual SLAM based on feature is Orb-Slam2 [1], which has been tested by a monocular camera, stereo camera, or RGBD camera. But due to its sensitivity to initialization, ambient lighting, and texture features, its robustness is poor. On this basis, a tight coupling optimization algorithm of IMU and camera is presented, which improves its robustness and accuracy to a certain extent. The representative algorithms mainly include the Orb-Slam3 algorithm [2] and the Vins-Fusion algorithm [3].

On the other hand, the LiDAR-based method has higher robustness due to its insensitivity to initial values and illumination. With the emergence of representative LiDAR odometry and mapping algorithm LOAM [4], LiDAR SLAM has achieved considerable development. Its series mainly include lightweight LiDAR SLAM algorithm LeGO-LOAM [5], lightweight LiDAR SLAM algorithm with context scanning SC-LeGO-LOAM [6], fast LiDAR odometry and mapping algorithm F-LOAM [7], etc. However, due to the lack of observation information from other sensors, the above algorithms have the problems of localization drift and LiDAR degradation in the corridor environment. The fusion of multiple observations with multiple sensors can further improve the accuracy and robustness of SLAM. The tightly coupled LiDAR inertial odometry and the mapping algorithm LIO-SAM [8] achieved a localization method with high accuracy and strong robustness by combining GPS, IMU, and LiDAR information optimization. LVI-SAM [9], a tightly coupled LiDAR vision inertial odometry, adds a visual-inertial subsystem on the basis of LIO-SAM, which makes the whole SLAM system still work normally when the LiDAR degrades and further improves the stability of the system. Multi-sensor localization and environment map construction of mobile robots have become mainstream methods.

Although the LiDAR-based SLAM algorithm achieves high accuracy and strong robustness through fusion with IMU and other sensors, the following problems still exist. Firstly, the motion of the mobile robot distorts the movement of the point cloud, thus causing errors in the matching of the LiDAR point cloud. Secondly, in the indoor environment, GNSS observation information is lacking, and in the open and long corridor environment, point cloud features are few, so it is difficult to make a good match of the point clouds. Thirdly, despite the addition of IMU for fusion, it is difficult to restore the degraded LiDAR SLAM system to normal by IMU status update information alone.

In view of the above problems, we design a multi-sensor fusion localization method based on visual marker constraint. The main work includes the following:We design the pre-integration model of encoder and IMU respectively and combine it with the pose estimation from the second stage to improve the problem of LiDAR degradation to a certain extent.We design low-frequency visual marker odometry, which is optimized jointly with LiDAR odometry to further improve the problem of LiDAR degradation.In order to better solve the wheel slipping and LiDAR degradation problems, we design an algorithm that can make the optimization weight of encoder odometry and LiDAR odometry adjust adaptively according to yaw angle and LiDAR degradation distance respectively. Experiments show that the proposed algorithm is very effective.

## 2. Relate Work

Outdoor mobile robots can obtain their latitude and longitude coordinates through the Global Navigation Satellite System (GNSS), so as to achieve high-precision localization. GNSS signal is weak due to the building barrier in an indoor environment, so GNSS technology cannot be used for indoor mobile robot localization. Indoor mobile robots mainly use multi-sensor for fusion localization. The methods of multi-sensor data fusion mainly include filtering and optimization.

Filtering methods mainly include extended Kalman filter (EKF), untraced Kalman filter (UKF), error-state Kalman filter (ESKF), particle filter (PF), and so on. Qin et al. designed an ESKF, recursively corrected the estimated state by generating new features in each iteration, and proposed the Lins algorithm [10], so as to fuse 3D LiDAR with 6-axis IMU in a relatively robust way and obtain a real-time state estimator. Xu et al. fused LiDAR feature points with IMU data by using a tightly coupled iterative extended Kalman filter and proposed the Fast-Lio algorithm [11], which could operate stably in a fast-moving, noisy or chaotic environment and proposed a new Kalman gain formula to reduce computational load. Based on the Fast-Lio algorithm, the team proposed the Fast-Lio2 algorithm [12], which directly matched the original point cloud with the map and made full use of the subtle features of the environment to improve the localization accuracy. Meanwhile, the data structure of the incremental KD tree was proposed to maintain the map with higher computational efficiency. The original system has been improved in two aspects of accuracy and real-time. In order to improve the tracking speed of the Fast-Lio2 algorithm, Bai et al. proposed the Faster-Lio algorithm [13], which improved the traditional voxel and proposed the point cloud spatial data structure of the incremental voxel without using the two main data structures of strict K-nearest neighbour and complex incremental KD tree, thus improving the operation speed of the system. Liu et al. used a multi-state constraint Kalman filter to fuse visual observation and IMU pre-integration information and used Helmert variance component estimation to adjust the weight of visual features and IMU pre-integration, so as to realize the accurate estimation of camera pose [14]. Anastasios et al. proposed a localization algorithm based on an extended Kalman filter based on visual inertia fusion and mainly constructed a camera observation model to describe the geometric constraint of multiple cameras observing static features, enabling state estimation in a wide range of environments [15]. Wang et al. proposed an adaptive unscented Kalman filter algorithm for time-varying noise covariance to calculate the optimal weight of each sensor information to achieve better fusion, so as to obtain the optimal state estimation [16]. Bloesch et al. proposed a monocular vision inertial mileage calculation method, which directly used the pixel intensity error of the image block to track, and tightly coupled the tracking of multi-layer features with the extended Kalman filter to accurately estimate the pose of the robot [17]. Dai et al. proposed an adaptive extended Kalman filter to fuse the visual-inertial odometer and the GNSS system when visual feature points are few or the GNSS signal is blocked, so as to make up for their shortcomings [18]. Geneva et al. proposed OpenVINS open platform for academic and industrial research and used a manifold sliding window filter to fuse Aruco 2D code features to estimate camera pose [19]. Frida et al. used the change of magnetic field in the environment to make up for the cumulative drift of the odometer base on an extended Kalman filter [20]. Asghar et al. combined the results of the two map matching algorithms with the extended Kalman filter used by GPS and dead reckoning to estimate the attitude of vehicles relative to the map [21]. Yang et al. proposed the strategy of front-end iterative Kalman filter and back-end graph optimization, which improved the robustness and pose accuracy of the system to a certain extent [22].

The optimization method mainly constructs the sensor data fusion as a nonlinear least square problem and uses the Gauss-Newton method, Levenberg-Marquardt method, or Dogleg method to solve the nonlinear problem iteratively. In the field of vision-based multi-sensor fusion localization, Campos et al. used visual methods to fuse IMU pre-integration information on the basis of Orb-Slam2 [1] and proposed the Orb-Slam3 algorithm [2], which can run robustly in real-time in both small and large indoor and outdoor environments. Lukas Von added IMU on the basis of the DSO algorithm to solve the problem that the scale could not be estimated by monocular. The optimization function included visual residuals and IMU residuals and the pose of the camera was estimated by solving the optimal solution of the optimization function [23]. Qin et al. adopted a tightly coupled approach and proposed the Vins-Mono algorithm [24] to obtain high-precision vision-inertial odometry by integrating the pre-integral measurement of IMU with the visual features of a monocular camera. Subsequently, the team added observation information of stereo camera on the basis of Vins-Mono to achieve a more powerful Vins-Fusion algorithm [3] and realized observation information fusion of IMU, monocular camera, and a stereo camera, making the system more robust. In the field of LiDAR-based multi-sensor fusion localization, the Google team developed the Cartographer algorithm [25]. IMU and encoder odometry were fused to obtain a pose estimator at the front end of SLAM, providing an initial value for LiDAR odometry. The nonlinear optimization problem was constructed by matching the current frame point cloud information of LiDAR with a subgraph. The initial state estimation results were obtained and then the pose was further optimized by the branch and bound method at the back end to obtain more accurate localization results and realize more accurate 2D occupancy grid map construction. In the field of 3D LiDAR SLAM, Ye et al. proposed the Lio-Mapping algorithm [26], a tightly coupled LiDAR and IMU fusion method, which minimizes the cost function of LiDAR and IMU measurement and reduces the cumulative drift, so as to obtain more accurate LiDAR state estimation. However, the algorithm is difficult to ensure the real-time operation of the system because of the huge amount of computation in the back end. In order to improve the real-time performance of the LiDAR inertial system, Shan et al. proposed the LIO-SAM algorithm [5], which realized the tight coupling of LiDAR and IMU in the front end and provided an initial value for frame image matching in the back end and carried out joint optimization of GPS observation, loop feature and LiDAR odometry increment in the back end. In addition to ensuring the localization accuracy and real-time performance of the system, GPS observation information is added to improve the robustness of the outdoor complex environment. In order to further improve the robustness of the SLAM system, some researchers have integrated the LiDAR subsystem with the visual subsystem. Zhu et al. introduced the LiDAR data into the Orb-Slam2 by taking advantage of the unique characteristics of the LiDAR data to realize the integration of LiDAR odometry and visual odometry [27]. On the basis of LIO-SAM, Shan et al. integrated stereo camera, LiDAR, IMU, and GPS based on the image optimization method and introduced global attitude map optimization based on GPS and loop, which can eliminate cumulative drift. This method has high estimation accuracy and robustness for various environments [6]. Zhao et al. adopted an image-centered data processing pipeline, which combined the advantages of the loose coupling method and tight coupling method and integrated IMU odometry, visual-inertial odometry, and LiDAR inertial odometry [28]. Although the accuracy, real-time, and robustness of the above algorithm can meet the localization task of mobile robots in an environment with rich features, the LiDAR SLAM algorithm based on feature matching degrades in the corridor environment with few features, resulting in poor accuracy and even failure of localization task. Torres-Torreti et al. eliminated the accumulated drift of the odometer through artificial landmarks, so as to realize the robust positioning of insufficient feature points in the tunnel [29].

We can see the localization method based on a visual marker, encoder, and IMU fusion can work well in the environment of LiDAR degradation. In this article, we jointly optimize the visual marker observation information and the state prediction information from LiDAR, encoder, and IMU to obtain a robust estimation of the real-time state of the mobile robot.

## 3. Visual Marker-Aided LiDAR/IMU/Encoder Integrated Odometry

The Marked-LIEO framework proposed in this paper is shown in Figure 1, which consists of three parts. The first part is the processing of sensor input data, including encoder, IMU, LiDAR, and camera. The encoder pre-integration module constructs a model of incremental encoder odometry, providing the system with encoder constraints. IMU pre-integration module constructs a model of incremental IMU odometry, providing the system with IMU constraints. LiDAR data is used to build LiDAR odometry based on LOAM. Different from LOAM, the predicted value of the LiDAR pose comes from the IMU pre-integration module and encoder pre-integration module. Camera data is used to build a low-frequency observation model based on a visual marker. It provides an absolute observation for factor graph optimization and improves the accuracy of final pose estimation.

The second part is the optimization of the first stage. This stage performs joint graph optimization for encoder constraint, IMU constraint, and the second stage pose constraint, so as to obtain the pose estimation of the first stage. On the one hand, the pose estimation of the first stage can be used to filter LiDAR motion distortion, on the other hand, it can provide key frame pose prediction for the second stage. In this process, considering the problem of wheel slipping, we propose a strategy of adaptively adjusting the optimal weight of the encoder to improve the accuracy of the pose estimation of the first stage.

The third part is the optimization of the second stage. This stage carries out joint graph optimization for visual marker constraint and LiDAR odometry constraint. In view of the LiDAR degradation problem, the visual mark constraint and LiDAR odometry constraint are added to the factor graph for factor graph optimization, so as to improve the accuracy of pose estimation in the second stage. Moreover, we propose the solution to LiDAR degradation, adaptively adjustment of LiDAR odometry optimized weight, improving the precision and the robustness of localization in the corridor environment. After the pose estimation of the second stage is completed, on the one hand, the end pose estimation result is used for the optimization of the first stage. On the other hand, the historical frame feature point cloud is constructed as a feature point cloud map containing corner and point features according to the estimated pose from the second stage, so that the next frame point cloud can complete scan to map matching with feature point cloud map and construct new LiDAR odometry.

### 3.1. Encoder Pre-Integration

We construct the pre-integration model of the encoder according to the movement of the two-wheeled mobile robot in the 2D plane, as shown in Figure 2.

Where A and B are the states of the wheels at time *i* and time *j* respectively. m1 and m2 are the moving distance from time *i* to time *j* calculated by the left encoder and the right encoder respectively. *b*, *r* and *θ* represent the wheel spacing, the turning radius of the left wheel, and the yaw angle of the wheel center at time *i* respectively. Line AB represents the moving distance of the wheel center from time *i* to time *j* and bisects the yaw angle of the wheel at time *i* and time *j*.

According to the above motion model, the encoder pre-integration can be expressed as follows:(1)$$\left\{ \begin{array}{l} \Delta \theta_{ij}=(m_{2}-m_{1})/b\\ \Delta x_{ij}=m_{ij}\cos(\Delta \theta_{ij}/2+\theta )\\ \Delta y_{ij}=-m_{ij}\sin(\Delta \theta_{ij}/2+\theta ) \end{array}\right.$$

The above encoder pre-integration realizes the estimation of the position and yaw angle of the mobile robot. Considering the wheel slipping and error of measurement deviation, we design the residuals between encoder pre-integration and LiDAR odometry from time *i* to time *j* as the encoder constraint factor, as shown in the following formula:(2)$$\left\{ \begin{array}{l} \delta p_{ij}=p_{L}-T_{LO}p_{O}(x_{ij},y_{ij})\\ \delta R_{ij}={\left(R_{LO}R_{O}(\theta_{ij})\right)}^{T}R_{L} \end{array}\right.$$
where $p_{L}$ and $p_{O}(x_{ij},y_{ij})$ are the displacement increment of the LiDAR odometry and the encoder odometry from time *i* to time *j* respectively. $R_{L}$ and $R_{O}(\theta_{ij})$ are the rotation increment of the LiDAR odometry and the encoder odometry from time *i* to time *j* respectively. $T_{LO}$ and $R_{LO}$ are the transformation matrix and rotation matrix from encoder frame to LiDAR frame respectively, which are obtained through offline calibration. $\delta p_{ij}$ and $\delta R_{ij}$ are the residuals of displacement increment and rotation increment from time *i* to time *j* respectively. Then the nonlinear least square function is constructed according to the above residuals and the optimal pose is acquired through the factor graph.

### 3.2. IMU Pre-Integration

The theoretical pre-integration result of rotation increment $\Delta R_{ij}$, velocity increment $\Delta v_{ij}$, and displacement increment $\Delta p_{ij}$ of IMU from time *i* to time *j* are as follows:(3)$$\left\{ \begin{array}{l} \Delta R_{ij}=R_{i}{}^{T}R_{j}=\prod_{k=i}^{j-1}\exp[(w_{k}-b_{g,k}-\eta_{g,k})\Delta t_{ij}]\\ \Delta v_{ij}=R_{i}^{T}(v_{j}-v_{i}-g\Delta t_{ij})=\sum_{k=i}^{j-1}\Delta R_{ik}(a_{k}-b_{a,k}-{ɳ}_{a,k})\Delta t_{ij}\\ \Delta p_{ij}=R_{i}{}^{T}(p_{j}-p_{i}-v_{i}\Delta t_{ij}-\frac{1}{2}g\Delta t_{ij}{}^{2})=\sum_{k=i}^{j-1}\left[\Delta v_{ik}\Delta t_{ij}+\frac{1}{2}\Delta R_{ik}(a_{k}-b_{a,k}-{ɳ}_{a,k})\Delta t_{ij}{}^{2}\right] \end{array}\right.$$
where $R_{i}$ and $R_{j}$ respectively represent the rotation matrix from IMU frame to world frame at time *i* and time *j*. $w_{k}$, $b_{g,k}$ and $\eta_{g,k}$ respectively represent angular velocity measurement value, angular velocity zero deviation, and angular velocity measurement noise at time *k*. $v_{i}$ and $v_{j}$ represent the velocity at time *i* and time *j* respectively, $a_{k}$, $b_{a,k}$ and ${ɳ}_{a,k}$ respectively represent acceleration measurement value, acceleration zero deviation, and acceleration measurement noise at time *k*. $p_{i}$ and $p_{j}$ represent the displacement at time *i* and time *j* respectively. *g* is the gravitational acceleration and its actual value is related to the geographical location.

Considering that the measurement of IMU is affected by gravity and noise and the zero bias of the gyroscope and accelerometer are not constant, we introduce the first-order Taylor approximation of the pre-integration results to the zero bias and obtain the pre-integration measurement results as follows:(4)$$\left\{ \begin{array}{l} \Delta {\hat{R}}_{ij}\approx \Delta R_{ij}\exp(\delta \phi_{ij})\exp(\frac{\partial \Delta R_{ij}}{\partial b_{g,i}}\delta b_{g,i})\\ \Delta {\hat{v}}_{ij}\approx \Delta v_{ij}+\delta v_{ij}+\frac{\partial \Delta v_{ij}}{\partial b_{g,i}}\delta b_{g,i}+\frac{\partial \Delta v_{ij}}{\partial b_{a,i}}\delta b_{a,i}\\ \Delta {\hat{p}}_{ij}\approx \Delta p_{ij}+\delta p_{ij}+\frac{\partial \Delta p_{ij}}{\partial b_{g,i}}\delta b_{g,i}+\frac{\partial \Delta p_{ij}}{\partial b_{a,i}}\delta b_{a,i} \end{array}\right.$$
where $\Delta {\hat{R}}_{ij}$, $\Delta {\hat{v}}_{ij}$ and $\Delta {\hat{p}}_{ij}$ respectively represent the measured values of rotation, velocity, and displacement increment from time *i* to time *j* after considering zero bias update. $\delta \phi_{ij}$, $\delta v_{ij}$ and $\delta p_{ij}$ respectively represent the measurement noise of rotation, velocity, and displacement, which can be approximated as Gaussian distribution. $b_{g,i}$ and $b_{a,i}$ respectively represent the zero bias of gyroscope and accelerometer at time *i*. Therefore, we can construct a residual function between the pre-integration theoretical value and the pre-integration measured value of IMU from time *i* to time *j* as follows:(5)$$\left\{ \begin{array}{l} \delta \Delta R_{ij}=\log(\Delta {\hat{R}}_{ij}{}^{T}\Delta R_{ij})\\ \delta \Delta v_{ij}=\Delta v_{ij}-\Delta {\hat{v}}_{ij}\\ \delta \Delta p_{ij}=\Delta p_{ij}-\Delta {\hat{p}}_{ij} \end{array}\right.$$
where, $\delta \Delta R_{ij}$, $\delta \Delta v_{ij}$ and $\delta \Delta p_{ij}$ respectively represent the rotation residual, velocity residual, and displacement residual of IMU from time *i* to time *j*. In order to make the IMU measurement results and the LiDAR odometry closely coupled, we set the rotation increment and displacement increment of the LiDAR odometry as the theoretical values of rotation increment and displacement increment of IMU respectively. Then we construct the nonlinear least square function according to the above residuals and optimize the optimal pose through the factor graph.

### 3.3. Marker-Based Observation

As a fixed symbol in the environment, a visual marker can be applied to the localization of mobile robots to improve the robustness of localization. In this paper, it is applied to the long corridor environment and the localization problem of long corridor LiDAR degradation can be effectively solved by jointly optimizing the visual marker constraint and the LiDAR odometry constraint. Next, we will introduce how to realize localization based on a visual marker.

Mainstream visual markers mainly include Rune-Tag [30], April-Tag [31], Chroma-Tag [32], Aruco-Tag [33], etc. Considering the positioning accuracy and recognition speed, we select Aruco-Tag as a visual marker. The core of localization based on a visual marker is to solve the PnP problem. Firstly, 2D pixel coordinates of the four corners of Aruco-Tag are obtained by a series of steps, including image segmentation, quadrilateral contour search, homography transformation, and Aruco dictionary decoding. Then, the 3D spatial coordinates of the four corners relative to the center of Aruco-Tag are calculated according to the size of Aruco-Tag. The relationship between 2D pixel coordinates and 3D spatial coordinates is as follows:(6)$$\left[\begin{array}{l} u_{i}\\ v_{i}\\ 1 \end{array}\right]=\frac{1}{z_{i}}{K(R}_{CM}\left[\begin{array}{l} x_{i}\\ y_{i}\\ z_{i} \end{array}\right]+t_{CM})$$
where $u_{i}$ and $v_{i}$ are the 2D pixel coordinates of the *i*-th corner point of Aruco-Tag relative to the camera. $x_{i}$, $y_{i}$ and $z_{i}$ are the 3D spatial coordinates of the *i*-th corner point of Aruco-Tag relative to the center of Aruco-Tag. $R_{CM}$ and $t_{CM}$ respectively represent rotation matrix and translation vector of Aruco frame to the camera frame. K is the internal parameter matrix of the camera. Since there is a deviation between the observation of pixel coordinates of Aruco-Tag corner points by the camera and the theoretical pixel coordinates of Aruco-Tag corner points after the transformation of space coordinates by internal and external parameters, we construct it as a nonlinear least square problem, where the optimization variable is the rotation matrix $R_{CM}$ and translation vector $t_{CM}$ from Aruco frame to the camera frame. The residual function is as follows:(7)$$\left\{ \begin{array}{l} {(R}_{CM}*,t_{CM}*)=\underset{{(R}_{CM},t_{CM})}{\arg\min}||\left[\begin{array}{l} u_{i}\\ v_{i}\\ 1 \end{array}\right]-\frac{1}{z_{i}}{K(R}_{CM}\left[\begin{array}{l} x_{i}\\ y_{i}\\ z_{i} \end{array}\right]+t_{CM})|{|}_{2}^{2}\\ T_{CM}*=\left[\begin{array}{cc} R_{CM}* & t_{CM}*\\ 0 & 1 \end{array}\right] \end{array}\right.$$
where $T_{CM}*$ Is the rigid transformation matrix from the Aruco frame to the camera frame. We use the Gauss-Newton method to obtain the nonlinear optimal solution $T_{CM}*$ iteratively. In order to obtain the transformation matrix of the mobile robot relative to the world frame, we define the frame relations of the localization system based on a visual marker, as shown in Figure 3.

In this paper, we set the LiDAR frame and robot base frame as the same frame, so the pose of the mobile robot in the world frame can be expressed as the transformation matrix $T_{WL}$ from the LiDAR frame to the world frame. As shown in Figure 3, $T_{WM}$ represents the transformation matrix from the Aruco frame to the world frame. $T_{MC}$ represents the transformation matrix from the camera frame to the Aruco frame. $T_{CL}$ represents the transformation matrix from LiDAR frame to camera frame. According to the chain rule, $T_{WL}$ can be obtained as follows:(8)$$T_{WL}{=T}_{WM}T_{MC}T_{CL}{=T}_{WM}{\left(T_{CM}\right)}^{T}$$
where $T_{CM}$ is obtained iteratively by the Gauss-Newton method. $T_{WM}$ and $T_{CL}$ are rigid connections between frames, which can be obtained by offline external parameter calibration. Finally, by fixing Aruco-Tag in the long corridor, we can obtain low-frequency observations based on a visual marker.

### 3.4. The First Stage Factor Graph Optimization

Considering that the second stage will construct a nonlinear least square problem when solving the LiDAR odometry and this problem is a nonconvex problem, the second stage is sensitive to the initial value in the iterative solution. The accurate pose estimation of the first stage can avoid the second stage falling into local optimization in the solution process. Therefore, the first stage optimally integrates the encoder constraint, IMU constraint, and the pose constraint from the second stage through factor graph optimization to achieve the fusion of encoder, IMU, and LiDAR measurement information, so as to obtain the pose estimation of the first stage. On the one hand, the pose estimation of the first stage is used for LiDAR point cloud distortion removal, on the other hand, it provides initial value prediction for the second stage. The factor graph of the first stage is shown in Figure 4.

In the actual system operation, the wheel may slip due to smooth ground, resulting in inaccurate yaw angle and deviation of overall localization. In order to improve the accuracy and robustness of the system, we propose when the yaw angle increases, the optimal weight of the encoder constraint decreases and approaches 0. Specifically, the optimized weight of the encoder is defined as follows:(9)$$w(M)=\frac{1}{||\theta ||+1},0<w(M)\leq 1$$
where w(M) is the optimized weight of the encoder and θ is the yaw angle of the mobile robot in the world frame. The experimental part of this paper will explain the precision and robustness of the first stage fusion localization can be improved by reasonable allocation of the optimal weight of the encoder.

### 3.5. The Second Stage Factor Graph Optimization

In order to obtain accurate pose estimation and solve the problems of cumulative drift and LiDAR degradation of LiDAR odometry, the second stage obtains relatively robust visual observation information based on a visual marker and integrates it with the LiDAR observation information, so as to improve the cumulative drift problem and improve the localization accuracy. Factor graph optimization is adopted in the fusion scheme to jointly optimize the visual marker constraint and LiDAR odometry constraint. The factor graph is shown in Figure 5.

In addition, in the process of factor graph optimization, in order to better solve the problem of the LiDAR degradation, we make an improvement on the optimization weight of the LiDAR odometry. LiDAR degradation problem refers to the mobile robot in the corridor environment, the feature points of the point cloud from the front and rear frame are similar, resulting in the Jacobian matrix of the residual function becoming a singular matrix, so the iterative solution fails and the localization deviates. Considering when the LiDAR odometry degrades, the localization result is different from the pre-integration result of IMU, and the greater the difference, the more serious the degradation is, we propose that adaptively adjust the optimization weight of the LiDAR odometry, as follows:(10)$$w(L)=\frac{1}{||d(L)-d(I)||+1},0<w(L)\leq 1$$
where w(L) is the optimized weight of the LiDAR odometry constraint. d(L) and d(I) represent the distance between the localization coordinates of the LiDAR and IMU and the origin of the world frame respectively. The experimental part of this paper will explain the accuracy and robustness of multi-sensor fusion localization can be improved by reasonable allocation of the optimal weight of the LiDAR odometry.

## 4. Experiment

In this paper, experiments are carried out from the perspectives of a simulation environment and a real environment. Sensor data in corridor environment, indoor environment, and outdoor environment are collected respectively to complete experiments in different scenarios, so as to verify the accuracy and robustness of the algorithm proposed in this paper.

Among them, the simulation environment is built by a Gazebo simulation platform, including an aluminum workshop, a corridor with a pavilion, and a pure corridor. The real environment includes an indoor long corridor and outdoor wall side. For the above two experimental environments, we use a simulated mobile robot and a real mobile robot to record the dataset of the corresponding environment respectively. The specific information is shown in Table 1.

In view of the above three experimental environments, we compare the proposed Marked-LIEO algorithm with ALOAM and LIO-SAM algorithms which can be regarded as algorithms without using visual markers. Due to the limitation of the experimental environment, the LIO-SAM algorithm as the comparison only uses the fusion odometry of IMU and LiDAR, ignoring the GPS module and loop closure module. Experimental results of the three algorithms in the above dataset verify the accuracy and robustness of the proposed algorithm in the corridor environment.

### 4.1. Simulation Environment

In this section, we conduct experiments through the simulation environment to verify the accuracy of the sensor fusion localization scheme proposed in this paper. Experiments are mainly conducted in two different simulation environments. The simulation environments are shown in Figure 6.

The simulated mobile robot uses an Ackermann steering structure and is equipped with a 16-line Velodyne simulated LiDAR, a 9-axis simulated IMU, and a monocular simulated camera. The sensor installation platform is shown in Figure 7. In addition, the positive direction of the robot is consistent with the *x*-axis direction of the corridor environment.

For ALOAM, LIO-SAM, and Marked-LIEO in the above two environments, the trajectories along the *x*-axis, *y*-axis, and *z*-axis with time are shown in Figure 8.

As shown in the figure above, the deviation of LOAM and LIO-SAM in the *z*-axis direction gradually increases with the increase of distance. This is because corner point features of the vertically distributed point cloud in the environment are few in the corridor environment, the constraint on the *z*-axis direction is poor. While Marked-LIEO with the adaptive optimization weight of LiDAR odometry makes the localization keep robust under the situation of the LiDAR degradation.

In order to conduct a quantitative error analysis on the localization algorithms, we perform trajectory error analysis on ALOAM, LIO-SAM, and Marked-LIEO, where the real trajectory is provided by the Gazebo simulation environment. First, the absolute pose error of ALOAM, LIO-SAM, and Marked-LIEO trajectories in the above two environments are solved respectively. Then, the overall deviation of the trajectory is counted by the root mean square error, as shown in Table 2.

As can be seen from the above table, the root mean square error of Marked-LIEO is smaller than that of LOAM and LIO-SAM, indicating that Marked-LIEO with the adaptive optimization weight of LiDAR odometry has a high localization accuracy in the corridor environment without LiDAR degradation.

### 4.2. Real Environment

In this section, we conduct experiments on the Outdoor wall side to verify the accuracy of the sensor fusion localization scheme after adding the adaptive optimization weight scheme of the encoder odometry. The test environment is the national intelligent network vehicle test area (Changsha). The experimental platform is a mobile robot equipped with a 16-line Velodyne LiDAR, a 9-axis HFI-A9 IMU, an Intel Realsense D435i depth camera, two encoders, a set of Asensing inertial navigation system, and a laptop with AMD R7 4800H 2.9GHz processor as shown in Figure 9. In addition, the positive direction of the robot is consistent with the *x*-axis direction of the corridor environment.

We compare ALOAM, LIO-SAM, and Marked-LIEO with the adaptive optimization weight of encoder odometry in the above outdoor wall side environment, as shown in Figure 10.

As can be seen from the figure above, due to the lack of IMU observation information, the localization accuracy of the ALOAM algorithm is the worst, with large drift in *x*, *y*, and *z-axis* directions. Although the LIO-SAM algorithm has a small deviation in the *x*-axis and *y*-axis directions, it produces a large drift in the *z*-axis direction. The Marked-LIEO algorithm combined with adaptive optimization weight of encoder odometry can better constrain the drift of the LiDAR odometry in all directions and has strong localization accuracy.

Then, quantitative experiments are carried out to evaluate the deviation between the localization trajectory of the above three algorithms and the real trajectory. Using the localization result of RTK as the real trajectory, we calculated the absolute pose error of LOAM, LIO-SAM, and Marked-LIEO respectively, and finally make statistics of the error results through the root mean square error, as shown in Table 3.

As can be seen from the above table, the root mean square error of Marked-LIEO is smaller than that of LOAM, and LIO-SAM, indicating that Marked-LIEO with the adaptive optimization weight of encoder odometry has a high localization accuracy and strong robustness in the outdoor wall side environment.

## 5. Conclusions

In this paper, a multi-sensor fusion localization algorithm based on visual marker constraint is proposed. Mainly based on visual marker constraint, the observation information of IMU, encoder, LiDAR, and the camera is integrated to realize state estimation of mobile robot in an indoor long corridor environment. The pose estimation of the first stage is obtained through the joint graph optimization of the odometry increment constraint of the encoder and IMU and the pose estimation constraint from the second stage. The pose estimation from the second stage is obtained through the joint optimization of the low-frequency visual marker odometry and LiDAR odometry. Aiming at the problems of wheel slipping and LiDAR degradation, we propose a strategy of adaptive adjustment of optimal weight of encoder odometry and LiDAR odometry. Finally, we realize multi-sensor fusion localization based on visual marker constraint by fusing measurement information of encoder, IMU, LiDAR, and camera. Nevertheless, the method proposed in this paper is not suitable for large and open outdoor environments, but more suitable for indoor long corridor environments lacking GNSS signal, including teaching buildings, aluminum workshops, hotels, and so on.

## Figures and Tables

**Figure 1 sensors-22-04749-f001:**
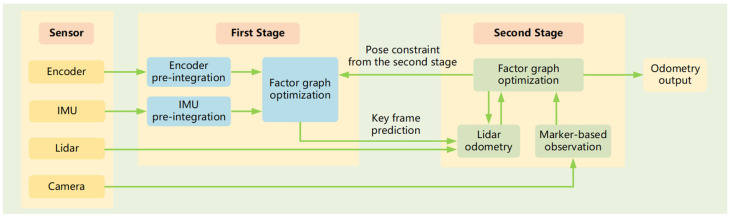
Marked-LIEO framework.

**Figure 2 sensors-22-04749-f002:**
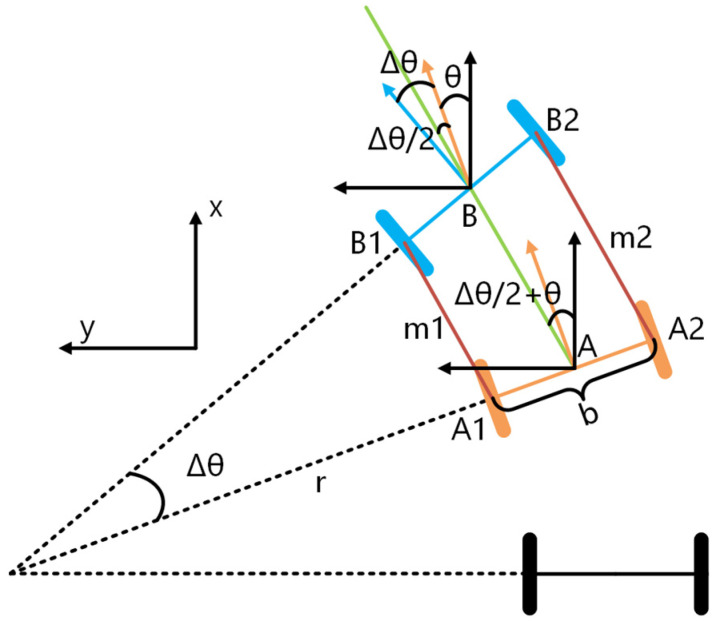
Motion model of two-wheeled mobile robot.

**Figure 3 sensors-22-04749-f003:**
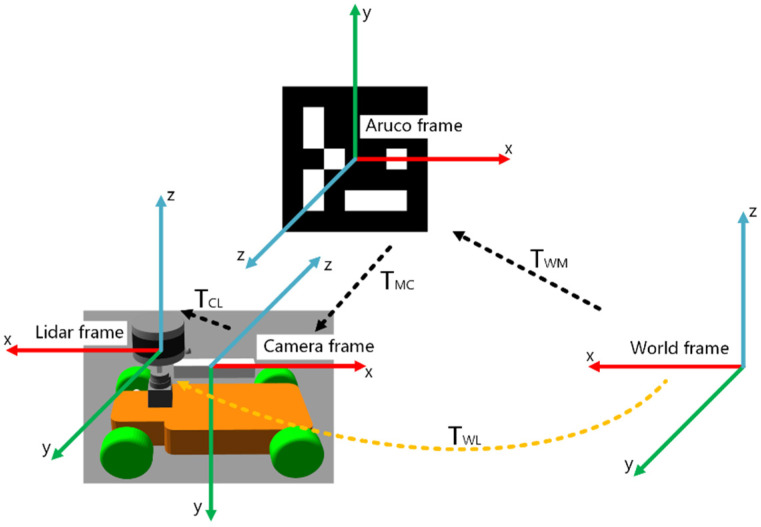
Localization system based on visual marker.

**Figure 4 sensors-22-04749-f004:**
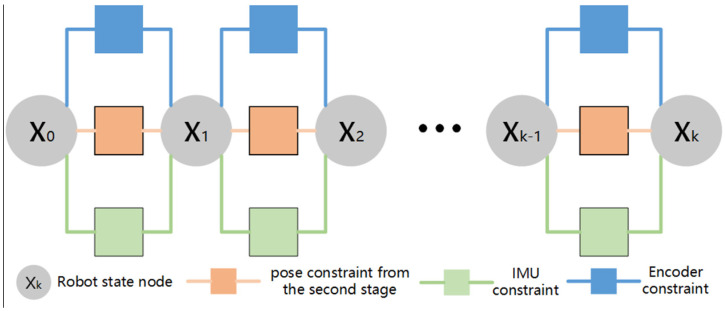
Factor graph in the first stage.

**Figure 5 sensors-22-04749-f005:**
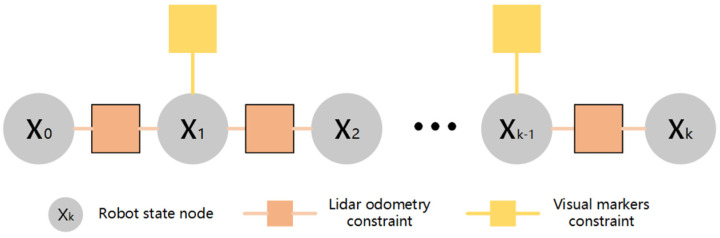
Factor graph in the second stage.

**Figure 6 sensors-22-04749-f006:**
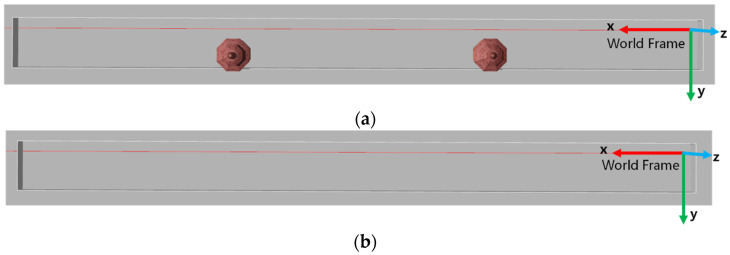
Simulation environment. (**a**) Corridor with pavilion; (**b**) Pure corridor.

**Figure 7 sensors-22-04749-f007:**
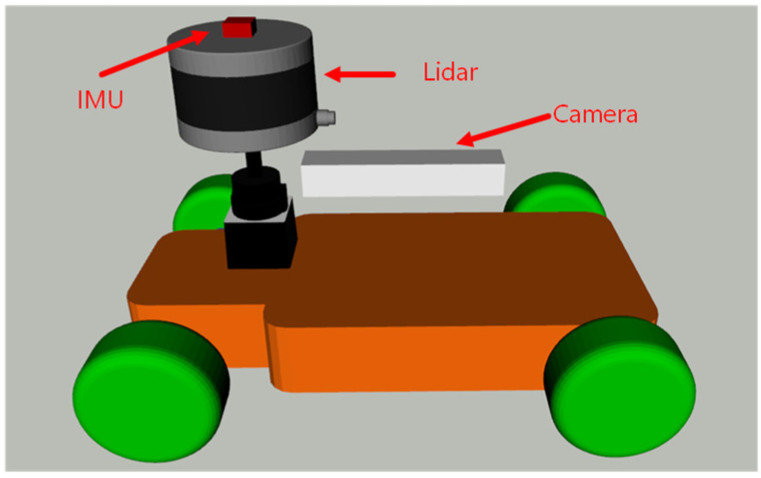
Sensor installation platform.

**Figure 8 sensors-22-04749-f008:**
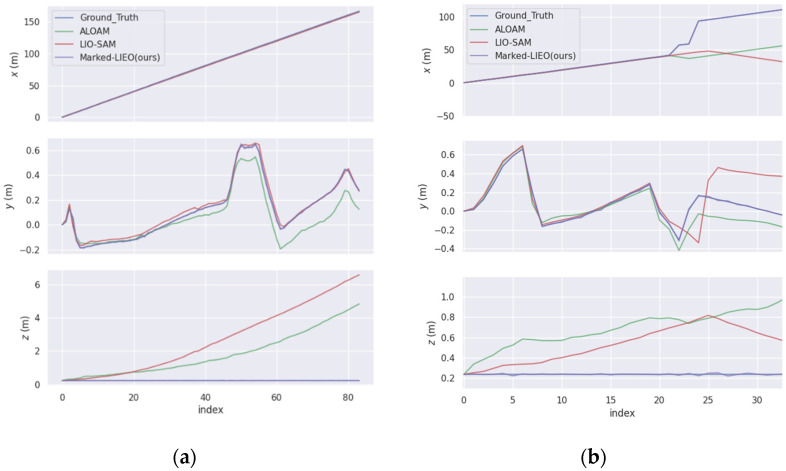
Three-axis trajectories of LOAM, LIO-SAM, and Marked-LIEO. (**a**) Three-axis trajectories in a corridor with a pavilion environment; (**b**) Three-axis trajectories in pure corridor environment.

**Figure 9 sensors-22-04749-f009:**
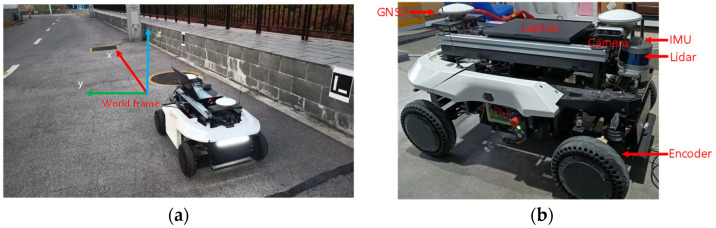
Experimental environment and sensor installation platform. (**a**) Outdoor wall side environment; (**b**) Mobile robot and sensor installation platform.

**Figure 10 sensors-22-04749-f010:**
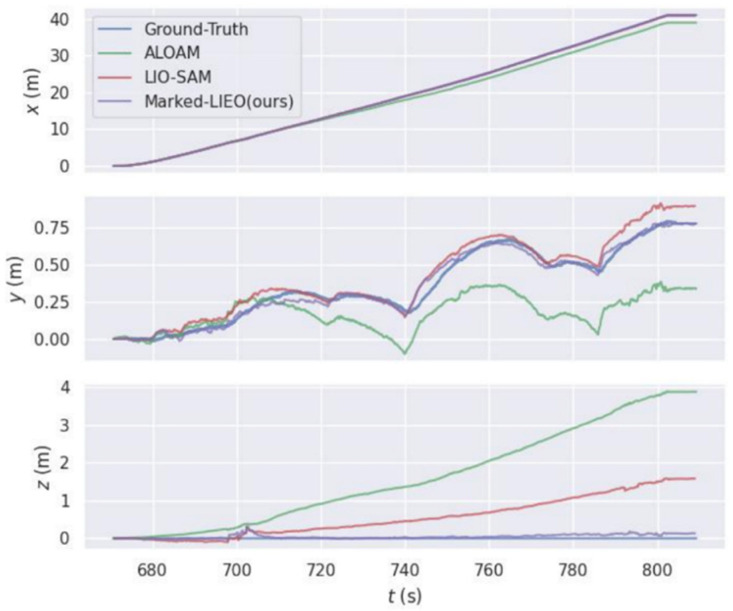
Three-axis trajectories of LOAM, LIO-SAM, and Marked-LIEO.

**Table 1 sensors-22-04749-t001:** Dataset Information.

Dataset	Category	Trajectory Length	Aruco-Tag Number
Corridor with pavilion	Simulation	165.56m	34
Pure corridor	Simulation	80.08m	16
Outdoor wall side	Outdoor	43.591m	11

**Table 2 sensors-22-04749-t002:** RMSE relative to ground truth. Bold indicates the best result under the same experimental conditions.

Dataset	ALOAM	LIO-SAM	Marked-LIEO (Ours)
Corridor with pavilion	2.69 m	3.64 m	**0.23 m**
Pure corridor	30.71 m	24.89 m	**0.34 m**

**Table 3 sensors-22-04749-t003:** RMSE relative to ground truth. Bold indicates the best result under the same experimental conditions.

Dataset	ALOAM	LIO-SAM	Marked-LIEO (Ours)
Outdoor wall side	2.40 m	0.78 m	**0.12 m**

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
