# Peer review of "Marked-LIEO: Visual Marker-Aided LiDAR/IMU/Encoder Integrated Odometry"

_sensors, 2022, doi:10.3390/s22134749_

Round 1

Reviewer 1 Report

OVERVIEW
This paper describes a fusion approach to compute robust odometry from visual markers, 2D lidar, IMU and encoders, and designed specially to overcome the degradation of performance in corridor-like environments.
The paper splits the problem in two-stages, each one involving a graphical-based optimization. The first stage involves encoder and IMU pre-integration, as well as uses the pose constraints from the second stage. The second stage uses the first stage output as predictions (or initial guess) and solves with constraints imposed bylidar and visual markers.

MAJOR COMMENTS
The paper presents a two stage approach to an optimization problem, and this should be better argued in the paper. Why it is better than a centralized optimization considering all constraints?

WORDING
Front-end and Back-end are used in this paper to call each optimization stage. As far as the reviewer knows, front-end is usually used in the SLAM/odometry context to the required signal processing, feature extraction and data asscoiation techniques to mount the optimization problem. The back-end is the optimization problem itself and the solving. To avoid confusion and for a better alignment with the existing literature, I'd suggest to rename these two optimization stages.

MINOR COMMENTS
- First paragraph misses some references on visual SLAM/odometry
- Section 3 should be reviewed since there is an explanation about "two parts" , but then inside the first part, there are "four parts". Please review the wording for better readability-
- In section 3.2, IMU measurements are considered directly rotation and velocities, but "Inertial" refers to rotation rate and accelerations. So it should be clarified at this point what are the raw measurements, and if a magnetometer is being used with an internal fusion at device level to provide rotation and velocity. 
- In equation 6, matrix sizes do not match. If T_CM encodes a full transformation as stated in the text, then it is an homogeneous matrix 4x4, K is 3x3 (typically), and the vectors are 2x1 and 3x1. Please write this equation more accurately. 
- T_CM is the unknown of the optimization problem. PLease specify if during the optimization, this variable is forced to be in a the rigid transformation manifold. 
- THe reviewer is curious about how lidar-camera extrinsics calibration has been computed. 
- In equation 9, which is the range of the theta angle ? [-pi, pi] ?
- In Figure 6, please indicate the world frame for each environment
- When describing the experiments, please indicate how the robot frame is initially placed with respect to the corridor. From the equation 8, is can be deduced that X is the corridor direction, but it is something that should be explicitly stated. 
- As far as the reviewer has understood, the comparison is with methods that do not use markers, so it is expected that a marker-aided approach performs better. 

Reviewer 2 Report

Overall, the article was interesting and easy to follow. I think readers will be impressed with the results from the comparisons with other algorithms.

There are a few specific suggestions:

1. When first using an Acronym put the words and then the Acronym in parentheses. This was done in some parts of the paper, but not in others.

2. When citing a list of authors, use last name only of first author followed by et al.

3. LiDAR is an Acronym itself and should not be lidar.

4. Lines 182-183 is a one sentence paragraph. This should be combined with paragraph below.

5. Lines 184-202 This paragraph is very confusing. The words first part are used several time for different things. I think this needs to be rewritten to provide better understanding.

6. Lines 283 and 286. Numbers below 10 should be spelled out (I believe this is the rule for most journals). 

7. Lines 327-330 This sentence does not make sense and has too many parts separated by commas. Split into several sentences for understanding.

8. Line 373, what are the Five environments? There are only 3 in the table and I could not figure them out from the paragraph above the table.

Round 2

Reviewer 1 Report

Most of the major comments of the initial review have been addressed. 

Still, the reviewer has three comments: 

1. In subsection 4.1, the authors claim: "considering the measurement information of the encoder is difficult to obtain in the simulation environment". But usually joint position and speed is available in common simulators , and can be directly translated to absolute or incremental encoder measurement respectively. 

2. Caption of figure6 seems to be incorrect: a) pure corridior, b) with pavilion

3. The arguments about why a decentralized way (solving two separate graph-based optimizations) is better than solving a single centralized graph with all constraints is not well explained, according two the reviewer point of view. 

4. The reviewer still thinks that "front-end" and "back-end" wording is used in this paper in a different way that most of the slam community does. A good reference for that can be encountered at the first section of [1]. Please consider this wording to better align with the community. 

[1] Cadena, Cesar et al. “Past, Present, and Future of Simultaneous Localization and Mapping: Toward the Robust-Perception Age.” IEEE Transactions on Robotics 32.6 (2016): 1309–1332.
